# Nonlinear Stress-Strain Model for Confined Well Cement

**DOI:** 10.3390/ma12162626

**Published:** 2019-08-17

**Authors:** Yan Li, Yunhu Lu, Ramadan Ahmed, Baoguo Han, Yan Jin

**Affiliations:** 1College of Petroleum Engineering, China University of Petroleum, Beijing 102249, China; 2Mewbourne School of Petroleum and Geological Engineering, The University of Oklahoma, Norman, OK 73019-1003, USA; 3School of Civil Engineering, Dalian University of Technology, Dalian 116024, China

**Keywords:** cement, stress–strain model, compressive strength, confining pressure, nonlinearity and ductility, mechanical properties

## Abstract

The cement sheath is the key for providing the zonal isolation and integrity of the wellbore. Oil well cement works under confining pressure, so it exhibits strong nonlinear and ductile behavior which is very different from that without confining pressure. Therefore, for the accuracy of the simulation and the reliability of well construction design, a reliable compression stress–strain model is essential for confined well cement. In this paper, a new axial stress–strain model for confined well cement is developed based on uniaxial and triaxial test data, examinations of failure mechanisms, and the results of numerical analysis. A parametric study was conducted to evaluate and calibrate the model. The model is simple and suitable for direct use in simulation studies and well design. Results from this study show the nonlinear compressive behavior of confined well cement can be predicted using the traditional uniaxial compressive strength test measurements.

## 1. Introduction

It is increasingly important to develop safer, environmentally friendly, and cost-effective technologies to satisfy the growing oil demand while conventional resources are decreasing [1]. As the main barrier to ensure zonal isolation and maintain the integrity of the wellbore, cement sheath must have an acceptable mechanical strength to withstand severe loading conditions resulting from the application of these new technologies. Consequently, more accurate mechanical strength models and better assessment of cement integrity are necessary to reduce the risk of failure. Numerical simulation is commonly applied to investigate the mechanical integrity of cement [2], also by using some software such as ABAQUS [3,4], ANSYS [5], and ADINA [1]. In all types of applications, an accurate well cement stress–strain model is required.

The linear elastic stress–strain model of hardened cement paste is universally used as a material function to perform numerical simulations [6]. Well cement is a brittle material when it is unconfined. Thus, using a linear elastic model is appropriate because cement fails quickly once the stress reaches the ultimate strength limit. For example, a linear elastic model can be used to simulate uniaxial tests of cement specimens. However, when well cement is set in the annulus and confined between casing and formation, it exhibits mechanical properties which are greatly different from that of unconfined cement [7]. Under confining pressure, the ultimate strain and strength of cement increase significantly, and its behavior becomes more ductile and nonlinear [6,8,9,10]. The peak volumetric strain and its corresponding axial strain increase linearly with increasing confining pressures [11,12]. The cement shows more plasticity and can hold more pressure cycles as the confining pressure increases [13]. Obviously, strong nonlinear characteristics cannot be described by a linear elastic stress–strain model. Moreover, the nonlinearity becomes stronger with the use of strength-enhancing additives [14].

In fact, compressive strength and the corresponding strain, ultimate strain, and post-peak behavior required to describe the nonlinearity of confined cement are difficult to obtain from triaxial test data [4], particularly those monotonically increasing curves shown in Figure 1. In addition, the post-peak material behavior of well cement is less reported, even for simple stress states [4]. Some investigations on the material behavior of confined concrete (i.e., a cementitious material) can be referred to due to the similarities of mechanical properties between cement and concrete. They both show brittleness without confinement and nonlinearity and ductility under confining conditions, though they have different properties. Confined concretes are classified based on the types of confinement as: Steel-confined concrete, steel tube confined concrete, and fiber-reinforced plastic (FRP) confined concrete. All of these confining conditions are similar to the entrapment of cement between the casing and formation. 

The stress–strain model of Popovics [15] and Mander et al. [16] was originally proposed for steel-confined concrete, and subsequently used in FRP confined concrete [17,18], and then adopted for steel tube confined concrete [19]. This model uses a single equation to relate the compressive strength of confined concrete to the unconfined concrete compressive strength and confining pressure. The corresponding strain is determined based on the ratio of confined compressive strength to unconfined compressive strength, and the strain at failure during the unconfined test. In addition, Mander’s model and ACI 318-08 models were also applied for confined geopolymer concretes [20,21,22,23,24]. 

Sakino et al. [25] proposed a similar stress–strain model for steel tube-confined concrete. The model is adopted from earlier studies [26,27,28,29]. A single equation is used to express the stress–strain model. The strength of confined concrete is determined by multiplying the strength of unconfined concrete by a confinement factor. The strain at failure is related to its mechanical properties and dimensions of the steel tube. The enhancement of the strain at failure due to confinement increases with the reduction of the diameter to thickness ratio of a steel tube. 

Another model [30] is developed for FRP-confined concrete. In this model, the nearly linear portion of the stress–strain curve is characterized by its slope and intercept. The model is appropriate for sufficiently confined concrete because it represents a monotonously increasing stress–strain curve. Other existing models [16,25] represent a decreasing post-peak stress–strain curve. In summary, using these single-equation models can represent the behavior of confined concrete. Though the models seem simple, they lead to a more complex equation form and complicated stress analysis procedures [31].

The typical bilinear stress–strain model, which includes two straight lines was applied for confined concrete [27,28]. This approach is simple, but straight lines cannot represent the nonlinear characteristic of the stress–strain curve. Therefore, a nonlinear function, such as a parabola, is commonly used to describe the pre-peak portion of the stress–strain model. Popovics’ curve was applied [32,33] followed by exponential function [34] or equations established using experimental data. Parabolas are directly used [35] or applied with some modification such as Hognestad’s parabola [31,36]. Other studies [19,37] used a parabola followed by a single or multiple straight lines. The parabolic function is also incorporated into some codes of practice such as BS 8110 [38] and EN 1992-1-1 [39] for concrete design.

In this study, the stress–strain model of confined cement is proposed based on experimental results, failure mechanism analyses, and trial calculations. The model can reasonably predict the compressive strength of cement under confining pressure. It describes the nonlinear characteristics of the material, which is not captured using the linear model. The proposed model was used in commercial software to carry out simulation studies and the predicted results are compared with experimental data to calibrate the model and evaluate its performance. 

## 2. Experimental Study

To calibrate and evaluate the new model, uniaxial, and triaxial compressive tests were conducted on conventional oil well cement specimens. Class G cement, silica flour, and additives were used to prepare test specimens. The compositions of all materials in the slurry formulation are presented in Table 1. Sample series 2 and 3 are conventional neat cements, which were obtained from the China National Petroleum Corporation (CNPC) and a previous study [40], respectively. All samples presented in Table 1 were tested using a triaxial rock testing system (GCTS Co. Model RTR-1000).

For specimens in Sample 1 series, one was tested in uniaxial (Sample 1-0), second (Sample 1-10), and third (Sample 1-20) in triaxial compression with 10 and 20 MPa confining pressure. These specimens were prepared with conventional cement and toughness-modifier was added during the slurry preparation. Therefore, the samples represent toughness-enhanced cement. The cement slurries were prepared according to API RP 10B-2 [41] standards. For Samples 1, the slurries were poured into cylindrical molds and cured for 7 days in 70°C water under atmospheric pressure. The size of the cylindrical samples was 2.54 cm in diameter and 5.08 cm in height (i.e., the height-diameter ratio was 2). This size is commonly used in standards and cement studies [14,42,43]. The height to diameter ratio of 2 meets applicable ISRM, ASTM, and cement testing standards [43], which is considered to be the accepted specimen geometry to test the compressive strength of cement [42,44,45,46,47,48]. Mechanical properties of cement Samples 1 in the uniaxial and triaxial tests were measured using an RTR-1000 triaxial rock testing system (GCTS Co., Tempe, AZ, USA). During the triaxial tests, confining pressure was first applied with the ramp rate of 50 N/S until 10 MPa for Sample 1-10 and 20 MPa for Sample 1-20. Then, an axial strain ramp rate of 0.05% per minute was maintained while measuring the deviator stress.

In Sample 2 series, two specimens were tested in uniaxial (Sample 2-0A and Sample 2-0B) and the other two specimens (Sample 2-10A and Sample 2-10B) in triaxial compression with 10 MPa confining pressure. For Sample 3 series, Sample 3-0 was used in uniaxial and Sample 3-20 in a triaxial test with 20 MPa confinement. The preparation of cement specimens and compressive strength test procedures were both according to API RP 10B-2 [41] and SY/T 6466 [43] standards. They were also 2.54 cm in diameter and 5.08 cm in height. 

Figure 1a shows the schematics of stress loads applied on specimens during uniaxial and triaxial compressive tests. The surrounding temperature was approximately 22°C for Samples 1 and 2, and 60°C for Sample 3. Figure 1b–d presents the stress responses in the compression experiments, i.e., deviator stress (y-axis) as a function of axial strain (x-axis). Deviator stress is equal to *σ*_1_-*σ*_3_. *σ*_1_ is the axial stress and *σ*_3_ is the confining stress. It can be seen that both the deviator stress and the ultimate strain are significantly enhanced under confining conditions. In addition, the enhancement increased with the level of confinement in Figure 1b.

## 3. Stress–Strain Model of Confined Well Cement 

Based on experimental results and extensive modeling studies on well cement, a stress–strain model is developed to represent the deformation behavior of cement under confined condition (Figure 2). The model considers three parts (segments) in the stress–strain curve with various parameters that are dependent on cement properties and confinement characteristics. Segment OA is represented by a second-degree parabola with point A at *ε_c_*_0_ and *f_c_*_0_. The *f_c_*_0_ represents the ultimate compressive strength of confined well cement. Peak stress segment (line AB) is a horizontal line, in which *ε_c_*_0_ and *ε_c_*_1_ are the minimum and maximum strains that correspond to the ultimate strength, *f_c_*_0_. The descending segment (BD) represents the post-peak (softening) region.

### 3.1. Modeling the Ascending Segment

Since many researchers and standards [19,35,37,38,39] consider that the appropriate approach for describing the first portion of the stress–strain curve is to use a modified parabola with various parameters. Hence, the ascending portion is modeled as a quadratic function. Thus:(1)$$\sigma =-\alpha_{1}\frac{f_{c0}}{\varepsilon_{c0}^{2}}\varepsilon^{2}+\beta_{1}\frac{f_{c0}}{\varepsilon_{c0}^{}}\varepsilon ,\varepsilon \leq \varepsilon_{co},$$ where *σ* and *ε* are the axial stress and axial strain, respectively. *α*_1_ and *β*_1_ are model parameters which control the slope and curvature of segment OA. The parameters are dependent on elastic modulus (*E)* and the stress at the linear limit (*σ_p_*_0_)_._ The parameters *E*, *σ_p_*_0_, *f_c_*_0_, *ε_c_*_0_, *α*_1,_ and *β*_1_ need to be obtained from a uniaxial test to fully describe stress–strain characteristics of cement in this segment.

#### 3.1.1. Determination of Elastic Modulus E and Stress at the Linear Limit σ_p0_

Elastic modulus affects mechanical properties of cement during the elastic deformation phase. The comparisons of the uniaxial and triaxial test data shown in Figure 3 demonstrate that the linear elastic phase of confined cement is basically close to that of unconfined cement. Particularly, the elastic modulus of confined cement is very close to that of unconfined one. For the Sample 2 series, the strain trends of uniaxial and triaxial tests are very similar when the axial strain is less than 0.25% (Figure 3d). This indicates that within the elastic limit, mechanical properties are little affected by confinement and the confinement effect is triggered once micro-cracks develop in the cement due to excessive loading. Therefore, in this study, elastic modulus *E* and the stress at the linear limit *σ_p_*_0_ of confined cement are considered to be the same as that of unconfined cement.

It can be seen from the uniaxial test results (Figure 4) that the stress–strain curve is basically linear when the stress is below 70% of *f_c_*_0_ and begins to deviate from the straight line when it is between 70% to 90% of *f_c_*_0_. Therefore, the stress at the linear limit (*σ_p_*_0_) of confined cement is expected to be between 0.7 *f_c_*_0_ and 0.9 *f_c0_*. If *σ_p_*_0_ = *φf_c_*_0_, then *φ* can be taken as the range between 0.7 and 0.9.

#### 3.1.2. Determination of the Ultimate Strength f_c0_ and the Corresponding Strain ε_c0_

After the linear elastic phase, as the cement approaches its unconfined ultimate strength, micro-cracks develop and grow to the extent that the Poisson ratio can no longer describe the relationship between the lateral and axial strains. In addition, the confinement, which is a result of the formation and casing becomes the sole restraining mechanism against crack development and catastrophic failure. In fact, it is difficult to obtain the ultimate strength of confined cement from the triaxial test results because as the load increases the cement continues to maintain its strength and resist the loading due to the confinement (Figure 3). The monotonous increase in the stress occurs even though parts of the material have reached the ultimate strength and can maintain it. Based on the above analyses of failure mechanism, *f_c0_* and *ε_c_*_0_ in the stress–strain model of confined cement are considered to be equal to the ultimate strength and corresponding strain of unconfined cement, which can be obtained from uniaxial test data.

#### 3.1.3. Determination of α_1_ and β_1_

According to Equation (1), parameters *α*_1_ and *β*_1_ control the slope and curvature of the ascending segment (OA). The peak point A (*ε_c_*_0_, *f_c_*_0_) and the point at the linear limit (*φf_c_*_0_/*E*, *φf_c_*_0_) are both in the curve. Hence, substituting the coordinates of point A (*ε_c_*_0_, *f_c_*_0_) into Equation (1), the following equation can be obtained.

(2)$$-\alpha_{1}+\beta_{1}=1$$

Then, substituting point (*φf_c_*_0_/*E*, *φf_c_*_0_) into Equation (1), the following equation can be obtained.

(3)$$-\alpha_{1}\varphi {\left(\frac{f_{c0}}{E\varepsilon_{c0}}\right)}^{2}+\beta_{1}\frac{f_{c0}}{E\varepsilon_{c0}}=1$$

Therefore, the expressions for *α*_1_ and *β*_1_ can be obtained combining Equations (2) and (3). Thus:(4)$$\alpha_{1}=\frac{\frac{E\varepsilon_{c0}}{f_{c0}}-1}{1-\frac{\varphi f_{c0}}{E\varepsilon_{c0}}},$$
(5)$$\beta_{1}=\frac{\frac{E\varepsilon_{c0}}{f_{c0}}-\frac{\varphi f_{c0}}{E\varepsilon_{c0}}}{1-\frac{\varphi f_{c0}}{E\varepsilon_{c0}}}$$

### 3.2. Modeling Peak Stress Segment

The strain increases rapidly after the peak point A, but the cement continues to resist the loading rather than immediately failing. A horizontal straight line or slightly ascending straight line is used in confined cementitious materials to describe the peak stress portion [31,38,39]. Besides this, some studies reported strain doubling in this phase [37,49]. Hence, this study assumes the peak stress portion (AB) to be a horizontal line, which extends from *ε_c_*_0_ to 2*ε_c_*_0_ (Figure 2). The equation for the peak stress portion is described as follows:(6)$$\sigma =f_{c0},\;\varepsilon_{c}0_{\;}\leq \;\varepsilon \leq \;{\varepsilon_{c}}_{1}$$ where *ε_c_*_0_ and *ε_c_*_1_ are the minimum and the maximum strain of the peak stress portion and *ε_c_*_1_ is equal to 2*ε_c_*_0_.

### 3.3. Modeling Descending Segment

The descending segment BD exhibits post-peak behavior. A number of studies [19,37,50] used straight lines to model the descending segment, which is simple and suitable for designing and performing numerical simulations. Therefore, in this study, the straight-line approach is adopted to describe the descending portion of the stress–strain model. Point C is necessary because experiments often do not prolong beyond this point due to the sudden failure of specimens. It would be better not to set the stress at point C too low. In this study, it is considered to be 0.65*f_c_*_0_. The strain at point C, *ε_c_*_2_, and the slope of BC are required to formulate an equation for this portion of the curve. The equation of this portion (BC) can be expressed as:(7)$$\frac{\sigma -f_{c0}}{\varepsilon -\varepsilon_{c1}}=\frac{0.35f_{c0}}{\varepsilon_{c1}-\varepsilon_{c2}},\;{\varepsilon_{c}}_{1}\leq \varepsilon \leq {\varepsilon_{c}}_{2}.$$

Let the slope of portion BC be -*C_Z_Zf_c_*_0_, and the following equation can be obtained:(8)$$C_{Z}Z=\frac{0.35}{\varepsilon_{c2}-\varepsilon_{c1}}$$ where both *C*_z_ and *Z* need to be determined. Based on trial calculations, a reasonable value of *Z* is found to be 30 and that of *C*_z_ is found to be between 0.85 and 2.5. Considering *ε_c_*_1_ = 2*ε_c_*_0_ and applying Equation (8), *ε_c_*_2_ can be determined using the following equation:(9)$$\varepsilon_{c2}=\frac{0.35}{C_{Z}Z}+2\varepsilon_{c0}.$$

Beyond point C, the straight line can be expected to continue with the same slope until the cement completely fails. *f_r_* represents the residual strength, and *ε_r_* represents the corresponding strain. If the confinement is large enough, it can be assumed that the cement is completely crushed at this stage; thus, *f_r_* is close to zero. However, to avoid numerical instability in the model, a value close to zero is recommended for *f_r_*, instead of zero.

## 4. Results and Discussions

### 4.1. Model Evaluation

To evaluate and calibrate the proposed stress–strain model, simulation studies were conducted using commercial software (ABAQUS) and results are compared with experimental measurements. First, the uniaxial test of Sample 1-0 was simulated using ABAQUS to calibrate the calculation process. The software is used to reproduce the test data shown in Figure 5. It can be seen that the simulation results predominantly match the experimental measurements, indicating the applicability of the calculation process with ABAQUS.

Arjomand et al. [4] used a uniaxial test stress–strain curve as a stress–strain model of cement to simulate its confined stress–strain behavior using ABAQUS. This procedure was applied in this study to reproduce triaxial test results of Samples 1-10 and 1-20 (Figure 6). It can be observed that numerically obtained curves do not fit the triaxial test data, though they have reasonably reproduced uniaxial test measurements (Figure 5). The results demonstrate that the stress–strain model of confined cement is significantly different from that of unconfined cement. Therefore, it is essential to propose an accurate and suitable stress–strain model for confined well cement.

The new stress–strain model is used to simulate the triaxial test results of all cement samples. The related parameters of the stress–strain model during simulations are presented in Table 2. These parameters were obtained according to the procedures proposed in Section 3. Figure 7 shows the corresponding models, in which Samples 1 represent toughness-enhanced cement and Samples 2 and 3 plain cement. Compared to the plain cement samples, the toughness-enhanced ones have higher strength and ultimate strain, thereby exhibiting significantly increased toughness and ductility.

It can be seen from Figure 8, using the proposed models, the numerical simulation results closely match the experimental data for both types of cement samples, while using Mander’s model [16] does not provide good predictions. When the new model is used, the predicted and experimental curves at the initial stage are in good agreement indicating that the values of parameters *E*, *α*_1_, *β*_1_, and *φ* are reasonable. The curves in the plastic stage also match well, which demonstrates the suggestions about the peak stress portion and descending portion are also appropriate for confined cement. Therefore, the proposed stress–strain model, including the equations and parameters are reasonable and suitable for not only plain cement but also toughness-enhanced cement under confining conditions.

### 4.2. Effects of Model Parameters 

#### 4.2.1. Effects of Strength f_c0_

As discussed in Section 3, the strength of confined cement in the model, *f_c_*_0_, is equal to that of unconfined cement obtained from uniaxial test data. The peak stresses in the uniaxial tests were 60 MPa for Sample 1, 47.5 MPa for Sample 2, and 37.8 MPa for Sample 3. To investigate the effects of the *f_c_*_0_ on the accuracy of the model, a parametric study has been conducted. The trial values of *f_c_*_0_ are 57, 60, and 65 MPa for Sample 1-10 and 1-20, 45–50 MPa for Sample 2, and 35–40 MPa for Sample 3. Other parameters of the stress–strain model are presented in Table 2. Figure 9 shows the numerical and experimental results for all samples. The deviator stresses increase slightly with the value of *f_c_*_0_. In fact, all numerical results are close to the experimental values, as long as *f_c_*_0_ in the model is taken as a value near the strength of unconfined cement obtained from uniaxial tests. It indicates that the suggested value of *f_c_*_0_ provides stable and accurate simulation results.

#### 4.2.2. Effects of C_Z_

*C_Z_* controls the slope of the descending portion of the stress–strain model. The appropriate value of this parameter is between 0.85 and 2.5. The simulations were carried out to reproduce the test data of Samples 1-10 and 1-20. It can be seen from Figure 10a,b that *C_Z_* has some effects in predicting the last stage of the experiment. The larger the value of *C_Z_* is, the faster the end of the curve drops. Figure 10c,d also illustrates this trend. The best fit can be obtained with *C_Z_* = 1 for Sample 1-10, and *C_Z_* = 0.85 for Sample 1-20. Even if the formulation is the same for the two samples, but the confinement is different, slightly different model parameters are needed. In this case, their models are different in the descending portion. It demonstrates again that, for a given cement type, the model predictions are not very sensitive to the model parameters.

## 5. Conclusions 

A new three-segment stress–strain model is proposed for confined well cement based on experimental data and simulation results. The model consists of a parabolic segment ascending to the peak stress, horizontal segment with constant stress, and linear descending segment. By applying the model in commercial software (ABAQUS), nonlinear characteristics of cement under confining pressure are accurately simulated. The results show good agreement between predictions and measurements obtained from neat cement under confining conditions. In addition, the model is used to simulate the triaxial test data of toughness-enhanced cement. The results show good agreement between simulation results and experimental data, indicating the accuracy and suitability of the proposed model. The new model is simple and can be directly implemented in simulation software used in designing. It requires model parameters that are obtained from uniaxial test data and reasonable estimation of *C_Z_* from available triaxial test measurements obtained from the same cement formulation. The sensitivity analysis conducted varying model parameters demonstrates that, for a given cement formulation, the model predictions are not very sensitive to the model parameters.

## Figures and Tables

**Figure 1 materials-12-02626-f001:**
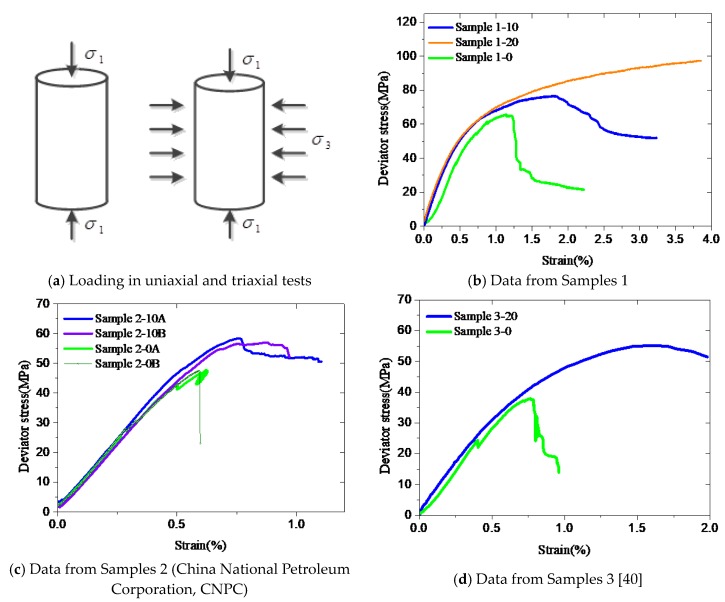
(**a-d**) Experimental results used for calibration of the model.

**Figure 2 materials-12-02626-f002:**
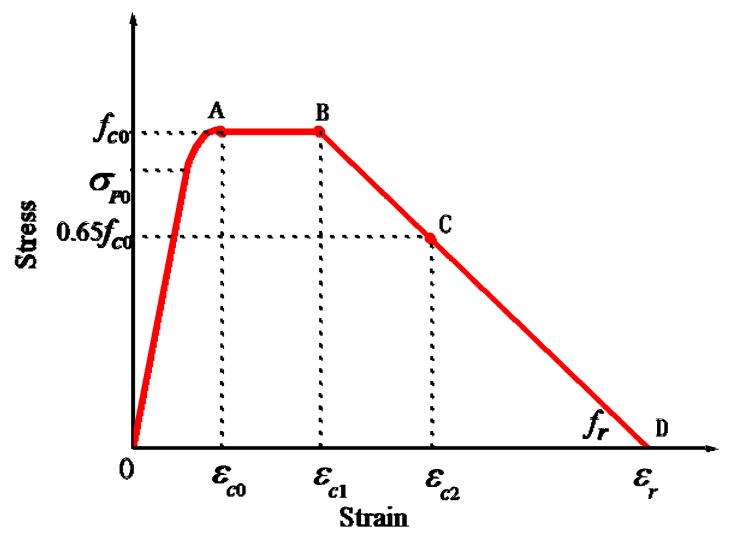
Proposed stress–strain model of confined well cement.

**Figure 3 materials-12-02626-f003:**
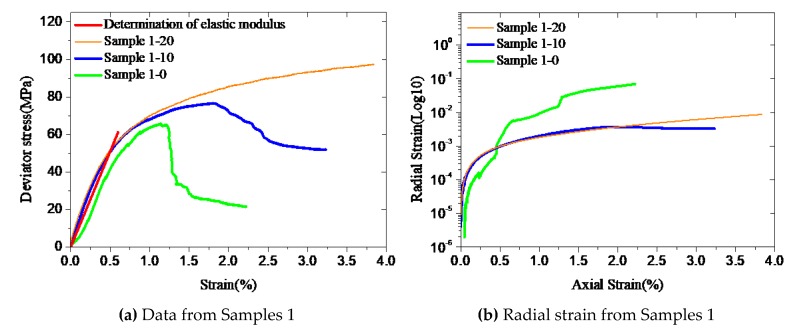
Determination of elastic modulus from test measurements (**a-e**).

**Figure 4 materials-12-02626-f004:**
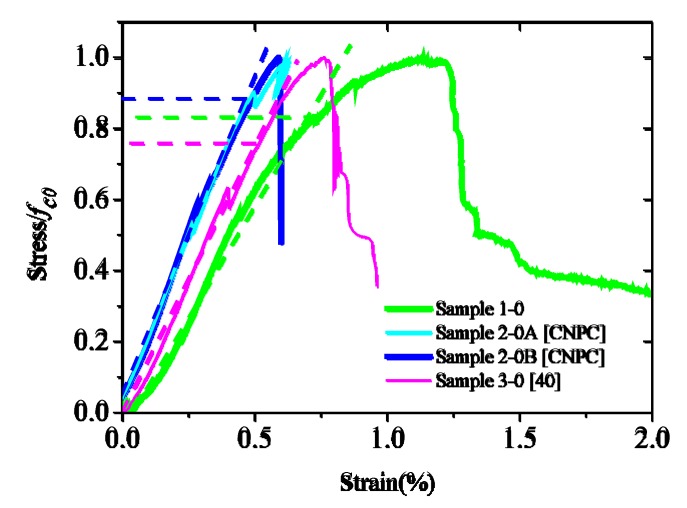
Determination of *σ_p_*_0_ based on uniaxial tests.

**Figure 5 materials-12-02626-f005:**
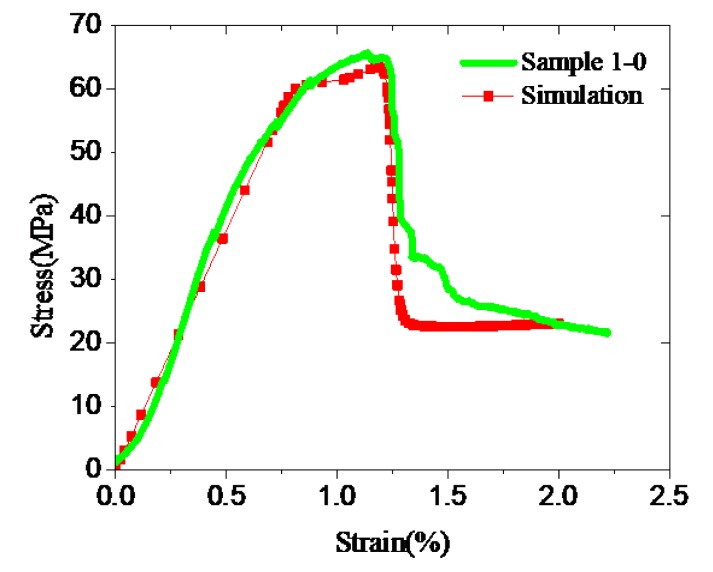
Simulation result and uniaxial test data of Sample 1-0.

**Figure 6 materials-12-02626-f006:**
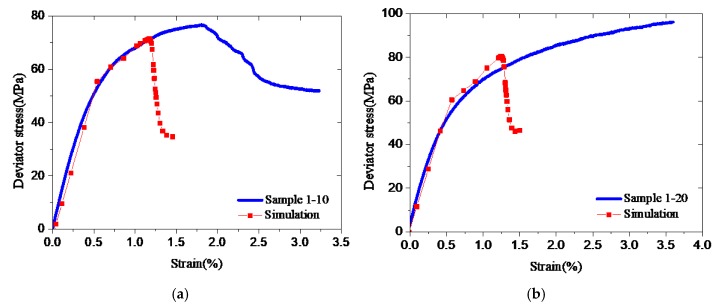
Simulations and triaxial test data for Sample 1. (**a**) Sample 1-10; (**b**) Sample 1-20.

**Figure 7 materials-12-02626-f007:**
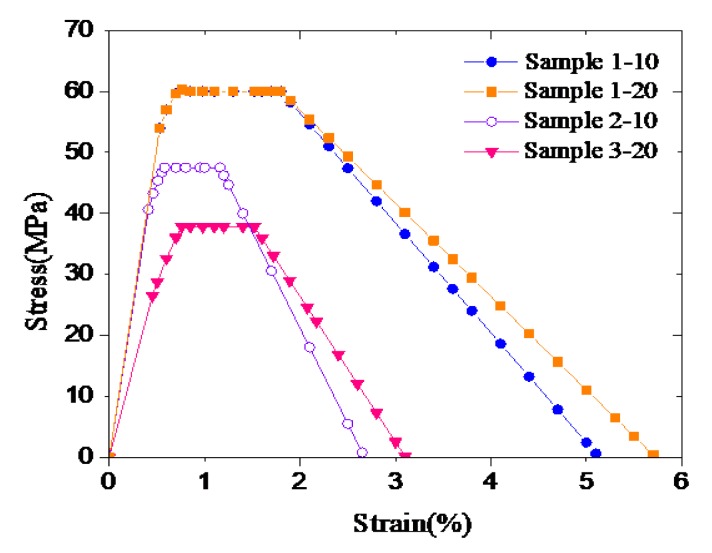
Stress–strain models of samples.

**Figure 8 materials-12-02626-f008:**
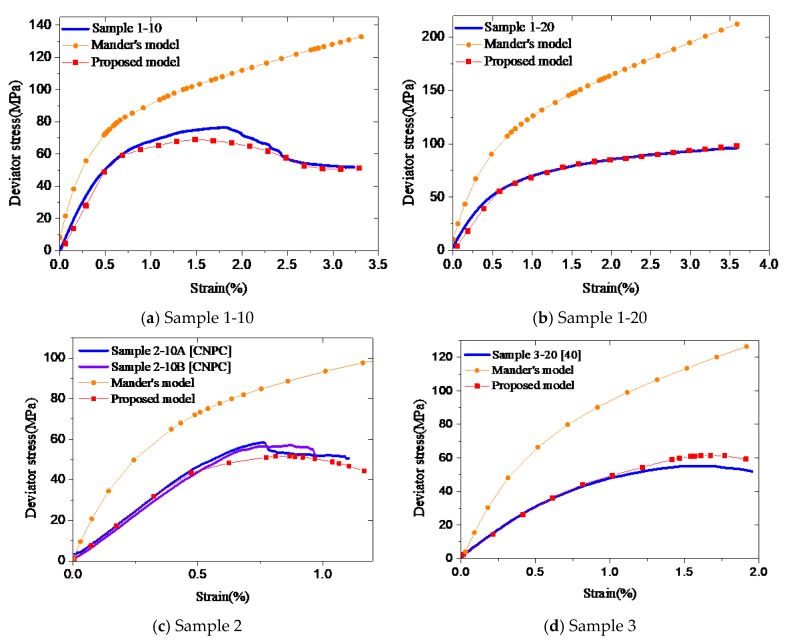
Comparison of numerical calculation with experiment results (**a-d**).

**Figure 9 materials-12-02626-f009:**
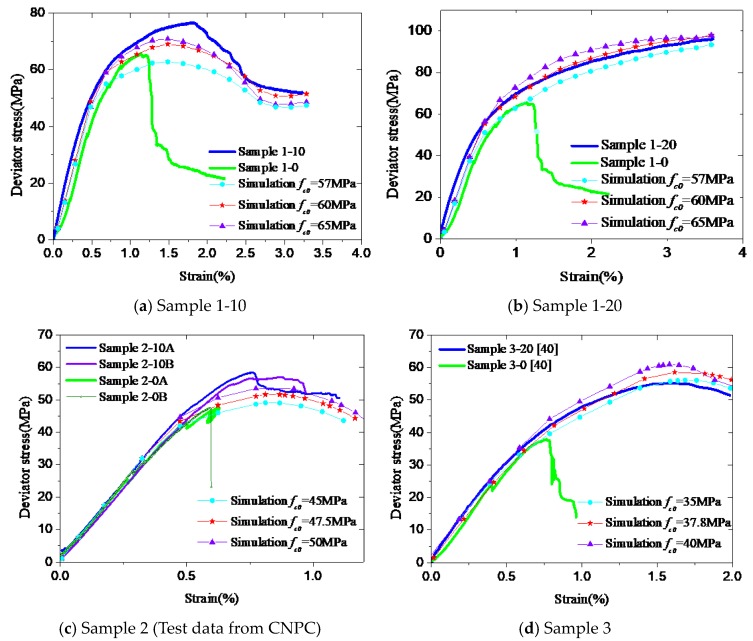
Effects of strength. (**a**) Sample 1-10, (**b**) Sample 1-20, (**c**) Sample 2, (**d**) Sample 3.

**Figure 10 materials-12-02626-f010:**
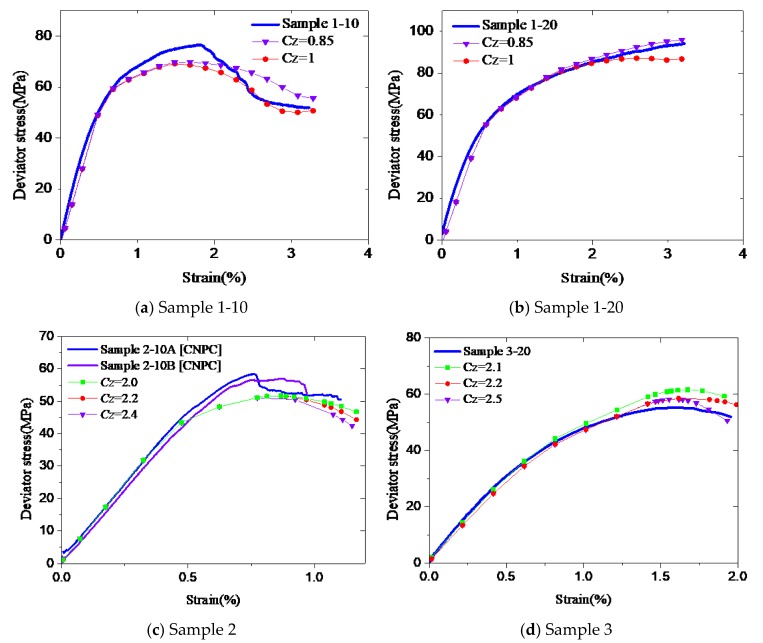
Effects of *C_Z._* (**a**) Sample 1-10, (**b**) Sample 1-20, (**c**) Sample 2, (**d**) Sample 3.

**Table 1 materials-12-02626-t001:** Compositions of all materials in the slurry formulation.

Sample	Compositions	Curing	Confinement [MPa]
Sample 1-0	Class G cement + water (w/c = 0.44) + silica flour + dispersant + antifoam agent + hydroxyethyl cellulose + latex modified mix	1 week at 70 °C and 0.1 MPa	0
Sample 1-10	Same as Sample 1-0	1 week at 70 °C and 0.1 MPa	10
Sample 1-20	Same as Sample 1-0	1 week at 70 °C and 0.1 MPa	20
Sample 2-0A	Class G cement + water (w/c = 0.44) + dispersant + stabilizing agent	1 week at 80 °C and 0. 1 MPa	0
Sample 2-0B	Same as Sample 2-0A	1 week at 80 °C and 0. 1 MPa	0
Sample 2-10A	Same as Sample 2-0A	1 week at 80 °C and 0. 1 MPa	10
Sample 2-10B	Same as Sample 2-0A	1 week at 80 °C and 0. 1 MPa	10
Sample 3-0 [40]	Class G cement + water (w/c=0.44) + dispersant + antifoam agent + hydroxyethyl cellulose	1 week at 60 °C and 20.7 MPa	0
Sample 3-20 [40]	Same as Sample 3-0	1 week at 60 °C and 20.7 MPa	20

**Table 2 materials-12-02626-t002:** Parameters of the model used in numerical simulation.

Cement	*E* [GPa]	Poisson’s Ratio	*φ*	*f_c_*_0_ [MPa]	*ε_c_*_0_ [%]	*C* _z_	Confinement [MPa]
Sample 1-10	10.2	0.2	0.85	60.0	0.848	1.00	10
Sample 1-20	10.2	0.2	0.85	60.0	0.848	0.85	20
Sample 2	9.65	0.24	0.88	47.5	0.580	2.20	10
Sample 3	5.85	0.24	0.70	37.8	0.760	2.20	20

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
