# Peer review of "Nonlinear Stress-Strain Model for Confined Well Cement"

_materials, 2019, doi:10.3390/ma12162626_

Round 1

Reviewer 1 Report

The manuscript presents the material model for the analyses of wellbore cement under triaxial compression. It is clearly written, the Introduction shows the author knowledge of the matter, however, I found some issues, that should be improved:

The formatting of the text is not in accordance with the requirements of Materials (references, equations etc)

I didn't find out how many specimens were tested in one series? Eg. two specimens as Sample 1, one tested in uniaxial and second in triaxial compression? This is important because testing of more specimens allows for statistical analysis and the elimination of erroneous results. In this context, I have doubts about cited Liu's research (Fig 1(c)). According to my knowledge degradation of modulus of elasticity is faster in uniaxial test (this is also confirmed by the results of the tests on Samples 1, 2 and 4). In Liu studies, modulus of elasticity even slightly increases above 50% of strength and is indeed approximately linear up to 90%. In addition, there is a large difference between the results of the uniaxial and triaxial tests. These are very unusual results, In my opinion their correctness needs to be verified (number of samples tested, repeatability of the result, tests with similar findings).

In the Conclusions authors wrote: …It requires model parameters that are obtained from uniaxial test data…, I can not agree with this statement. The Cz parameter was selected by matching the results of numerical calculations and triaxial tests. The authors have shown its large dispersion (between 0.85 and 2.5). The authors see the problem, as shown in the point 4.2.2, however, the sensitivity analysis of the model was provided for a small variation of Cz (eg for Samples 1 and to between 0.85 and 1.0 – why not for 2.5). In my opinion, the problem of the coefficient Cz needs to be discussed in more detail, especially its adoption without a triaxial test.

Page 4/line 9  …, Sample 3 was …  

Author Response

Dear Editor-in-chief,

We are grateful to the editors and reviewers for their time and constructive comments on our manuscript. We have implemented their comments and suggestions and wish to submit a revised version of the manuscript for further consideration in the journal. Changes in the initial version of the manuscript are highlighted in the revised version. We also provide a point-by-point response explaining how we have addressed each of the editors or reviewers’ comments. 

We look forward to the outcome of your assessment.

Yours sincerely,

On behalf of the co-authors

Yan Li

Reviewer 2 Report

This study presents a stress-strain model for well cement. The following sections should be revised:

1.    The reference study is not enough. Most of the references about models are 15 years ago.

2.    Please add chemical and physical compositions of materials used in this study

3.    Because Sample 2 is same as sample 1, please combine them as one sample

4.    I want the temperature and pressure of the test. For well cement, the temperature and pressure are much higher than ordinary cement

5.    The slope of figure 2 is not continuous at the joint section. Does it reasonable?

6.    Only 4 samples are used in this study. The number of samples is to limit. More samples are necessary.

7.    As shown in table 1, huge parameters are used to fit the results. Please express the parameters of the model as functions of cement properties, such as strength or etc.

8.    In section 4.1, authors use Abaqus program. What is the difference between your model and the inherent model of Abaqus?

9.    More verifications from other researchers’ study are necessary

Author Response

(The authors gave the same response as above.)

Author Response

(The authors gave the same response as above.)

Round 2

Reviewer 3 Report

Reviewer gratefully acknowledges  the efforts of authors made on the revised manuscript.